

# Downscaled surface mass balance in Antarctica: impacts of subsurface processes and large-scale atmospheric circulation

Nicolaj Hansen[1,2], Peter L. Langen[3], Fredrik Boberg[1], Rene Forsberg[2], Sebastian B. Simonsen[2], Peter Thejll[1], Baptiste Vandecrux[4], and Ruth Mottram[1]

[1]DMI, Lyngbyvej 100, Copenhagen, 2100, Denmark
[2]DTU-Space, Kongens Lyngby, Denmark
[3]Now at: iClimate, Department of Environmental Science, Aarhus University, Denmark
[4]Geological Survey of Denmark and Greenland, Copenhagen, Denmark

**Correspondence:** Nicolaj Hansen (nichsen@space.dtu.dk)

**Abstract.** Antarctic surface mass balance (SMB) is largely determined by precipitation over the continent and subject to regional climate variability related to the Southern Annular Mode (SAM) and other climatic drivers at the large scale. Locally however, firn and snow pack processes are important in determining SMB and the total mass balance of Antarctica and global sea level. Here, we examine factors that influence Antarctic SMB and attempt to reconcile the outcome with estimates for total

mass balance determined from the GRACE satellites. This is done by having the regional climate model HIRHAM5 forcing two versions of an offline subsurface model, to estimate Antarctic ice sheet (AIS) SMB from 1980 to 2017. The Lagrangian subsurface model estimates AIS SMB of 2473.5±114.4 Gt per year, while the Eulerian subsurface model variant results in slightly higher modelled SMB of 2564.8±113.7 Gt per year. The majority of this difference in modelled SMB is due to melt and refreezing over ice shelves and demonstrates the importance of firn modelling in areas with substantial melt. Both

the Eulerian and the Lagrangian SMB estimates are within uncertainty ranges of each other and within the range of other SMB studies. However, the Lagrangian version has better statistics when modelling the densities. There is a mean bias in modelled density of -24.0±18.4 kg m$^{-3}$ and -8.2±15.3 kg m$^{-3}$ for layers less than 550 kg m$^{-3}$ for the Eulerian and Lagrangian framework, respectively. For layers with a density above 550 kg m$^{-3}$ the bias is -31.7±23.4 kg m$^{-3}$ and -35.0±23.7 kg m$^{-3}$ for the Eulerian and Lagrangian framework, respectively. The mean firn 10 m temperature bias is 0.42-0.52 °C. Further, analysis of

the relationship between SMB in individual drainage basins and the SAM, is carried out using a bootstrapping approach. This shows a robust relationship between SAM and SMB in half of the basins (13 out of 27). In general, when SAM is positive there is a lower SMB over the Plateau and a higher SMB on the westerly side of the Antarctic Peninsula, and vice versa when the SAM is negative. Finally, we compare the modelled SMB to GRACE data by subtracting the solid ice discharge, and find that there is a good agreement in East Antarctica, but large disagreements over the Antarctic Peninsula.There is a large difference

between published estimates of discharge that make it challenging to use mass reconciliation in evaluating SMB models on the basin scale.





## 1 Introduction

The Antarctic Ice Sheet (AIS) has the potential to raise global sea level by 58 m (Fretwell et al., 2013) and it is therefore of

utmost importance to understand its role in present sea level change in order to project it into the future. At present the AIS contributes to sea level rise by 0.3±0.16 mm per year based on the average ice mass loss of 109±56 Gt per year between 1992 and 2017 (Shepherd et al., 2018). An accelerating mass loss has been observed in West Antarctica and over the Antarctic Peninsula (AP), in the last four decades (Forsberg et al., 2017; Rignot et al., 2019). In the light of this acceleration, climatic changes are of particular interest as such might induce ice sheet dynamical instability, by changing the mass influx to the ice

sheet. The ice sheet mass balance (MB) can be split in an atmospheric and ice dynamic component:

$$MB = SMB - D \ , \tag{1}$$

where D is the solid ice discharge in the form of iceberg calving, SMB is the surface mass balance composed of precipitation (P, snowfall and rain), sublimation and evaporation ($S$) from the surface and runoff (RO) of meltwater (SMB=P-S-RO). Of these components, precipitation is by far the largest contributor (Krinner et al., 2007), and consists primarily of snow at higher

altitudes while melt and runoff of surface melt is largely confined to ice shelves and elevations less than 1400 m above mean sea level (Bell et al., 2018). Sublimation and evaporation are however important across most of the continent due to low humidity and high wind speeds (Palm et al., 2017). If SMB < D the total mass balance is negative and the ice sheet loses mass, and thereby contributes to global sea level rise. Here we focus on the SMB component of the mass balance, to pin-point the immediate forcing to ice sheet dynamical instability.

Regional Climate Models (RCMs) are most often used to downscale coarser global models and reanalysis because they add further detail, due to their higher resolution, e.g in the mountainous areas where the climate can be affected from local orography creating katabatic winds or orographic forced precipitation (Rummukainen, 2010; Feser et al., 2011; Rummukainen, 2016). Mottram et al. (2020) evaluated the atmospheric output from five different RCM simulations of Antarctic SMB driven by ERA-Interim (1987-2017). These five models showed mean annual SMB ranging from 1961±70 Gt per year to 2519 ±118 Gt

per year. In the literature, individual model evaluations can be found which are within the same SMB range: COSMO-CLM[2]: 2177 ±80 Gt per year (Souverijns et al., 2019), MAR v3.6.4: 2200 ±115 Gt per year (Agosta et al., 2019), and RACMO2.3p2: 2219 ±109 Gt per year (Van Wessem et al., 2018). The overall model spread in SMB models corresponds to approximately 2 mm of sea level change per year. Mottram et al. (2020) also showed that when compared to in-situ observation from both automatic weather stations and glaciological stake measurements, the data availability proved insufficient to distinguish between

better performing model estimates. Fettweis et al. (2020) found similar conclusions for Greenland, where the RCMs showed different strengths and weaknesses when evaluated both spatially and temporally. Mottram et al. (2020) and Verjans et al. (2021) furthermore showed, that subsurface processes that drive melt and refreezing are extremely important when estimating the SMB. Hence, we here include firn processes by forcing a newly developed full subsurface SMB model for Antarctica with HIRHAM5 (Christensen et al., 2007) (1979-2017), to assess the effects of firn processes on estimates of ice sheet SMB. This

subsurface model accounts for the physical properties of the uppermost part of the AIS, including density and temperature and outputs the SMB.



Acknowledging that the in-situ observations might be challenged judging the performance of the SMB model (Mottram et al., 2020), we also compare our model results with a GRACE gravimetry estimate of the mass balance to determine any systematic biases. Finally, studies have shown that precipitation is not only the largest contributor to Antarctic SMB (Krinner et al., 2007; Agosta et al., 2019) it also has a spatial heterogeneous distribution varying over time, which affects the SMB (Fyke et al., 2017). Regional scale events like the heavy snowfall in Dronning Maud Land have an important measurable effect on AIS SMB (Lenaerts et al., 2013; Turner et al., 2019). Different representations of these may explain differences between modelled SMB e.g. (Mottram et al., 2020) as well as discrepancies between the GRACE mass balance and SMB-D solutions. Our study therefore also quantifies how regional climate indices affect SMB on basin scale.

Regional circulation patterns including ENSO (El Nino Southern Oscillation), the BAM (Baroclinic Annular Mode) and the Pacific- South American patterns (PSA1 and PSA2) have previously been identified as important determinants on weather and climate variability in Antarctica. However, empirical orthogonal functional analysis of southern hemisphere 500 hPa geopotential height (Marshall et al., 2017), demonstrates that the southern annular mode is the most important of these regional circulation indices and for this reason we concentrate on its effects in this study. Marshall et al. (2017) found that the phase of the southern annular mode (SAM), which describes pressure anomalies and precipitation in the southern hemisphere (Fogt and Bromwich, 2006), strongly affects the precipitation pattern over the AIS. The SAM is an atmospheric phenomenon found across the extratropical southern hemisphere that influences the climate over and around Antarctica (Fogt and Marshall, 2020). Studies have shown that the phase of SAM, can have great impact on the surface climate in Antarctica, such as the temperature (Thompson and Solomon, 2002; Van Lipzig et al., 2008), sea ice extent (Hall and Visbeck, 2002), pressure (Van Den Broeke and Van Lipzig, 2004) and especially precipitation (Van Den Broeke and Van Lipzig, 2004). A positive SAM giving more precipitation in West Antarctica but reduced precipitation over the Antarctic plateau. Therefore, we also investigate the spatial distribution of SMB over the grounded AIS (GAIS) in relation to the phase of the SAM.

## 2 Methods

### 2.1 HIRHAM5 Regional Climate Model

The HIRHAM5 RCM is a hydrostatic model with 31 atmospheric layers, developed from the physics scheme of the ECHAM5 global climate model (Roeckner et al., 2003) and the numerical weather forecast model HIRLAM7 (Eerola, 2006). HIRHAM5 has been optimised to model ice sheet surface processes that often are neglected or simplified in global circulation models, for full description we refer to Christensen et al. (2007) and Lucas-Picher et al. (2012). Here HIRHAM5 is forced at the lateral boundaries at 6-hourly intervals with relative humidity, temperature, wind vectors and pressure from the ERA-interim reanalysis (Dee et al., 2011), further, daily values for sea ice concentration and sea surface temperature are also used. HIRHAM5 calculates the full surface energy balance at the surface, based on model physics as described in Lucas-Picher et al. (2012); Langen et al. (2015); Mottram et al. (2017). Following, Langen et al. (2015) the albedo is derived by a simple broadband model ranging from 0.85 for fresh snow to 0.6 for exposed glacier ice. HIRHAM5 also calculates the amount of snowfall, rainfall, water vapor deposition and snow sublimation that occurs at the surface. Specific, for the HIRHAM5 Antarctic simulations,





was that we used the Antarctic domain defined in the Coordinated Regional Climate Downscaling Experiment (CORDEX) (Christensen et al., 2014) and downscaled it further to $0.11°(\approx 12.5$ km) spatial resolution with a dynamical time step of 90 seconds.

## 2.2 Subsurface model:

The subsurface model was originally built on ECHAM5 physics (Roeckner et al., 2003) but has been updated to include a

sophisticated snow and ice scheme and thereby updates the subsurface snow layers with snowfall, melt, retention of liquid water, refreezing, runoff, sublimation and rain (Langen et al., 2015, 2017). Thereby, the subsurface model is forced with the snowfall, rainfall, evaporation, sublimation and surface energy fluxes from HIRHAM5. These include net latent and sensible heat fluxes and downwelling shortwave and longwave radiative fluxes for 6-hourly intervals over the period 1979-2017. To reduce RCM spin-up effects, such as misrepresentation of the physically state of the atmosphere e.g temperature, the first year

is removed from the results. Despite the forcing is based on 6 hourly values the subsurface scheme is used to simulate the subsurface at 1-hour time steps. The horizontal resolution of the subsurface model is following the $0.11°$ native resolution of HIRHAM5.

As the Antarctic SMB may be sensitive to the subsurface model setup we here use two versions of the subsurface model (Langen et al., 2017). Common for both model version is their meltwater percolation, firn compaction and heat diffusion

schemes, meltwater in excess of the irreducible water content (Coléou and Lesaffre, 1998) is transferred from one layer to the next using a parameterization of Darcy flow developed by Hirashima et al. (2010) with hydraulic conductivity values calculated from Van Genuchten (1980); Calonne et al. (2012) and coefficients from Hirashima et al. (2010). The impact of ice content on the layer's conductivity is described by the parameterization by Colbeck (1975). When meltwater can infiltrate into a subfreezing layer, it is refrozen and latent heat is released. Firn density is updated at each time step for compaction under

each layer's overburden pressure using the parameterization by Vionnet et al. (2012).

However the two model versions differ in the management of the layers within the model. The first model version developed by Langen et al. (2017), has 32 subsurface layers with a fixed predefined mass, expressed in m water equivalent (w.eq.), given by $D_N = D_1 \lambda^{N-1}$, where $N$ is the given layer and $D_1$ = 0.065 m w.eq. This fixed model implies an Eulerian framework, meaning that mass is advected through layers of fixed mass. When snowfall occurs at the surface, it is added to the first layer

and an equal mass from that layer is shifted to the underlying layer. The same goes for each layer in the model column. The same procedure is followed when mass is removed from the top layer due to runoff or sublimation. Then each layer takes from their underlying neighbour an amount of snow/firn equivalent to the mass lost at the surface. The temperature and density of the layers are updated as the average between the snow or firn that is received by the layer, and what remains there. In the following we refer to this model version as the Fixed model.

The second model version uses a Lagrangian framework for the layer evolution developed by Vandecrux et al. (2018, 2020a, b). Layers evolve through splitting and merging dynamically based on a number of weighted criteria. This dynamical model, henceforth referred to as Dyn model, has 64 subsurface layers, which is fixed during the simulation. When snowfall occurs at the surface, it is first stored in a "fresh snow bucket". When this snow bucket reaches 0.065 m w.eq., its content is added as a



new layer at the surface of the subsurface scheme and two layers need to be merged elsewhere in the model column. The layer merging scheme assesses how likely a layer is to be merged with its underlying neighbour based on seven criteria: the layers' difference in temperature, density, grain size, water content, ice content, depth and the thickness of the layers. The first five criteria makes it preferable to merge layers with small differences, the sixth criterion, makes it preferable to merge deep layers rather than shallow layers, in this case the shallow layer limit is set to 5 m w.eq., this criterion carries twice the weight of the first five. The final criterion says that no layer can be thicker than a maximum thickness, in this case 10 m w.eq., this is set to avoid the deepest layers continuing to grow. A weighted average of the criteria, where the first five are weighted equally, while the depth and thickness criteria are weighed double and triple respectively, is used by the model to determine which layers should be merged. When surface sublimation or runoff occur it is taken from the snow bucket and then from top layer. When a layer decreases in thickness and its mass reaches 0.065 m w.eq., then it is merged with the underlying layer and another layer can be split in two elsewhere in the model column. The splitting routine is based on two criteria; thickness of the layer where thick layers are more likely to split; and shallowness where shallow layers are more likely to split. The two criteria are weighted 60/40. However, the minimum thickness of any layer is always 0.065 m w.eq. to avoid numerical instability.

The dynamic-layer model updates to the melting scheme also include simultaneously melting of ice and snow, when there is sufficient energy available, whereas in the Fixed model the surface snow melts first followed by ice. Furthermore, the Dyn models runoff is routed downstream using Darcy's law and the local surface slope, whereas the Fixed model follows Zuo and Oerlemans (1996) and excess water in a layer cannot be transferred to the underlying neighbour. Both the Fixed and Dyn versions require a fresh snow density value when adding snowfall at the surface. We here use the Antarctic parameterization from Kaspers et al. (2004) which use local climatological means of skin temperature, 10 m wind speed and accumulation rates.

### 2.3 Experimental Set-up

The Fixed model was initialized with a firn column with uniform density of 330 kg m$^{-3}$, and a temperature at the bottom of the firn pack given by the climatological mean of the HIRHAM5 2 m temperature. Spin-up was performed by repeating a decade (1980-1989) multiple times. The state of the subsurface at the end of each decade was used as the initial state for the next iteration. There were no appreciable shifts in the Antarctic climate from 1980-2019 (Medley et al., 2020), so the 1980s can be used as a representative decade for spinning up the subsurface. The Fixed subsurface scheme was spun-up over 25 iterations (250 years). Afterwards, the actual experiment ran from 1979-2017. To limit computing-time, the dynamical model was initialized with the last spin-up from the Fixed model and extrapolated to the 64 layers of the Dyn model. From then, additional spin-ups (1980-1989) ensured that the dynamical splitting and merging of layers had time to evolve throughout the firn pack. Two spin-up experiments have been carried out for the Dyn model; one that uses three decades of additional spin-up (Dyn03), resulting in a total of 280 spin-up years (250 from the fixed model and 30 year in the dynamical model), and one that uses 15 decades of spin-up (Dyn15), resulting in a total of 400 spin-up years.

All three model simulations (summarized in Table 1) provide outputs of monthly and yearly means of all 3d variables (density, grain size, firn temperature and ice/water/firn content), daily 2d fields (SMB, runoff, super imposed ice, melt, albedo, ground heat flux, refreezing, diagnosed snow depth, net short wave and net long wave radiation) of the surface variables,



**Table 1.** Model overview and main differences.

|  | Fixed | Dyn03 | Dyn15 |
|---|---|---|---|
| Thickness | Constant over time and space | Varies over time and space | Varies over time and space |
| #Layers | 32 | 64 | 64 |
| Spin-up [yr] | 250 | 280 | 400 |
| Melt | Fist snow then ice | Snow and ice simultaneously | Snow and ice simultaneously |

furthermore daily columns for specified coordinates interpolated to the nearest grid cell, have been retrieved, for comparison of in-situ measurements. For the two simulations with dynamical layer thickness, the daily 3d fields are interpolated into a fixed
grid, with the same number of layers, so time averages could be calculated.

## 2.4   Regional drivers and mass balance

The SAM is characterized in Fogt and Bromwich (2006) as the zonal pressure anomalies in the high southern latitudes having opposite sign to those of the mid latitudes. The SAM drives the westerly winds around Antarctica but the stream oscillates north-south. The SAM can have three phases: positive, neutral or negative, where positive creates a higher pressure over the
mid latitudes and lower pressure over Antarctica, and thus moves the westerly winds closer to Antarctica. A negative SAM creates a lower pressure over the mid latitudes and a higher pressure over Antarctica, moving the westerly winds north. When neutral there is no pressure difference anomaly. To investigate how the phase of SAM affects the SMB, monthly SAM data, as calculated by Marshall (2018), have been used. From 1980-2017, 261 months showed a positive SAM (SAM+), 193 months a negative SAM (SAM-) and 2 months were neutral, the SAM data are given as one monthly number, i.e one number for
the entire Antarctic domain. To see if there is a link between SAM and SMB, the monthly SMB values were divided into two groups: SAM+ and SAM-. Then the mean SMB for all months with SAM+ were subtracted from the mean SMB for the entire period and likewise for SAM-. To see if there was a statistically robust difference in the $\delta$SMB signals we performed a bootstrapping analysis, using 1200 random resamplings without replacement of the SAM data, to see if the $\delta$SMB signals could be replicated randomly, if it could be produced randomly the signal would not be robust. Statistically robustness has here
defined as $\delta$SMB valuas falling outside the 5-95[th] percentile range. In order to maintain the seasonal variability in the SMB, the SAM data were shuffled in sets of 12 – in this way the order of the months was maintained and thus the seasonal cycle retained. Then confidence intervals were determined as the 5[th] and 95[th] percentiles of the distribution of the resampled $\delta$SMB values.

   Observing the mass balance can be helpful to assess the spatial patterns of SMB and evaluate the modelled results. Mass
balance can be derived from gravimetric measurements from space. Here GRACE/GRACE-FO mass loss time series data were computed for the period 2002-2020, using a mascon approach based on CSR R6 level 2 data, complete to harmonic degree 96 (Forsberg et al., 2017). The lowest degree terms were substituted with satellite laser ranging data, and glacial isostatic adjustment corrections from the model of Whitehouse et al. (2012). From Eq. 1 we know that MB should be equal to SMB





minus discharge (SMB-D), so to evaluate our SMB model performance, GRACE and SMB-D have been plotted. The discharges
values were derived from two studies: Gardner et al. (2018) and Rignot et al. (2019). Gardner et al. (2018) gave values from
2008 and 2015 here we took the mean value and used as $D_{Gardner}$ over the period. Rignot et al. (2019) have derived decadal
mean discharge values from 1999-2010 and 2010-2017, for $D_{Rignot}$ the relevant discharge values were used. The SMB value
used here is for the grounded AIS only and since the modelled SMB values are quite similar over the grounded AIS, it is only
shown here for the Dyn15 simulation.

## 3  Results

In the model mean (1980-2017) of the three SMB simulations (Fig. 1a), we see that the majority of the total AIS (ToAIS) has a
positive SMB, only a few regions show a negative SMB: Larsen ice shelf, George IV ice shelf, coastal regions of Queen Maud
Land, the Transantarctic mountains, near Amery ice shelf and some coastal areas in East Antarctica. Near Vostock in East
Antarctica, the SMB is less than 25 mm w.eq. per year. The SMB increases towards the coast due to higher precipitation. The
highest SMB is greater than 2000 mm w.eq. per year and is found on the windward (western) side of the AP, whereas the most
negative SMB, -500 mm w.eq. per year, is found on the leeward (eastern) side of the AP (Fig. 1a). All the model simulations
show nearly identical SMB values over the GAIS, however they differ the most near the coast in West Antarctica and the AP
as Fig. 1b shows. Here, we see that the $\delta$SMB (model minus mean) show that the Fixed version has a higher SMB of up to 550
mm w.eq. over the Larsen ice shelf relative to the model mean, in Dyn03 the SMB values differ between -350 and 400 mm
w.eq. from the model mean, this change occurs over a few grid cells. In Dyn15 the SMB differ up to -650 mm w.eq. compared
to the model mean over the Larsen ice shelf. Since the Fixed is above the model mean, over the Larsen ice shelf, and Dyn15
is below the model mean, it looks like that the rapid change from negative to positive $\delta$SMB in Dyn03 over Larsen ice shelf is
due to lack of spin-up. Below the AP, of the coast of Ellsworth Land and Marie Byrd Land, the Fixed version models a lower
(-75 mm w.eq.) SMB than Dyn03 (35 mm w.eq.) and Dyn15 (50 mm w.eq.) all relative to the model mean. Around Alexander
Island in the Bellingshausen sea, both the Fixed and Dyn15 have a lower SMB compared to Dyn03. The differences in spatial
distribution show that in areas where melt occurs, the SMB is very sensitive to which subsurface scheme is used.





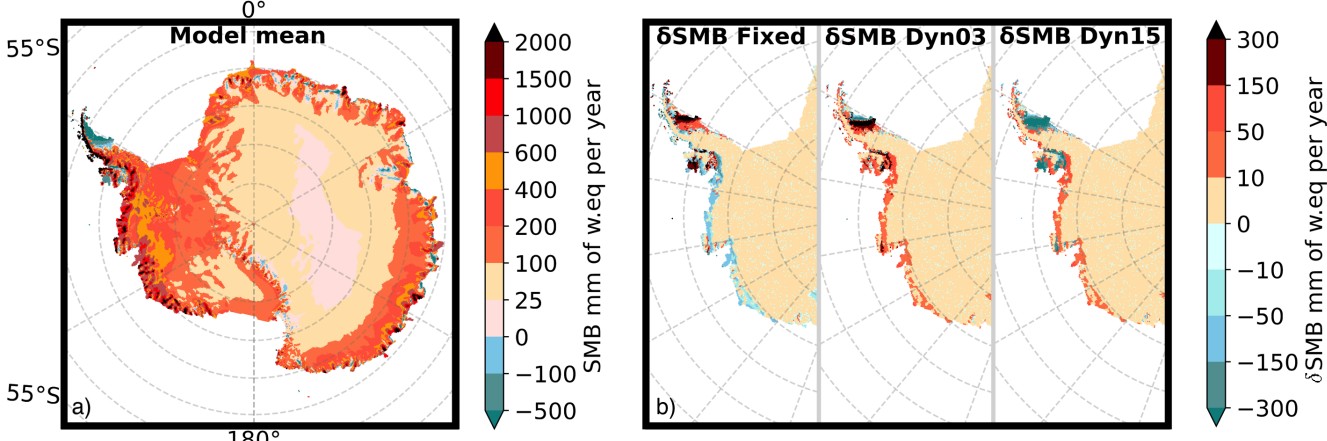

**Figure 1.** Mean SMB from 1980 to 2017 in mm per year of water equivalent. a) the mean of the model mean, note nonlinear colour bar. b) West AIS where the $\delta$SMB has the largest differences between model versions, (model minus ensemble mean).

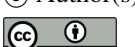



**Figure 2.** Integrated precipitation (A), SMB (B), melt (C), refreezing (D) and runoff (E) all in Gt per year, (F) show the runoff to melt fraction. For the three model simulations, for the entire AIS with ice shelves (ToAIS) and for the GAIS. Note different values on the y-axis.

The model differences are seen in the integrated values for precipitation, SMB, melt, refreezing and runoff, for both ToAIS and the GAIS (Fig. 2), and summarized in Table 2. As all model simulations are forced using the same precipitation field (Fig. 2a) and since the precipitation is the main driver of the SMB, the variability of the modelled SMB closely follows the precipitation variability. The spread in modelled mean melt, refreezing and runoff are respectively 1%, 11% and 8% smaller when including the ice shelves compared to only taking the GAIS, whereas the spread in mean SMB becomes 3% greater. To better compare the melt, refreezing and runoff from the different simulations, the fraction of runoff to melt is shown in figure 2f. Dyn03 has the smallest runoff fraction whereas Dyn15 and Fixed are quite close to each other. This implies, that even




**Table 2.** Yearly mean SMB, melt, refreezing, runoff, precipitation and runoff fraction (runoff over melt), ± with respective standard deviations, for both the total ice sheet (ToAIS) and the grounded ice sheet (GAIS). Note that all the model simulations are forced with the same precipitation.

| Model | | SMB [Gt yr$^{-1}$] | Melt [Gt yr$^{-1}$] | Refreezing[Gt yr$^{-1}$] | Runoff [Gt yr$^{-1}$] | Precipitation [Gt yr$^{-1}$] | Runoff fraction [%] |
|---|---|---|---|---|---|---|---|
| Fixed | ToAIS | 2564.8±113.7 | 695.3±132.4 | 463.7±97.3 | 208.3±47.5 | 2970.9±122.1 | 0.30±0.06 |
| | GAIS | 1995.2±95.7 | 180.0±49.5 | 125.1±40.3 | 48.8±10.4 | 2193.8±98.0 | 0.28±0.05 |
| Dyn03 | ToAIS | 2583.4±121.6 | 984.2±166.1 | 748.9±132.5 | 189.6±29.9 | — | 0.20±0.03 |
| | GAIS | 1995.4±99.3 | 247.7±61.7 | 215.3±54.1 | 48.6±7.0 | | 0.21±0.05 |
| Dyn15 | ToAIS | 2473.5±114.4 | 1004.5±173.7 | 674.5±121.7 | 299.5±47.1 | — | 0.30±0.03 |
| | GAIS | 1963.3±96.2 | 262.3±65.8 | 200.8±51.3 | 80.6±13.7 | | 0.32±0.05 |

though the magnitudes between the simulations are quite different, the refreezing capacity of the Fixed and Dyn15 are near
equal and Dyn03 has the smallest refreezing capacity. Note also that the melt is 289 and 309 Gt per year higher in the Dyn03
and Dyn15 respectively, compared to the Fixed model. Again this is focused largely over the ice shelves, and is most likely due
to their being more bare ice with a lower albedo in these simulations.

### 3.1 Evaluation against observations

Modelled firn densities are evaluated using the SumUp dataset (Koenig and Montgomery, 2019). When disregarding firn cores
shallower than 2 metres, there were 139 density profiles left (Fig. 3), all the references for the firn profiles can be found in
the reference list. These profiles vary in depth, from a few metres to 100 metres, but the majority are drilled to 10 metres
depths. Knowing the coring date, we compare it to the modelled density of the nearest grid cell the same date. Before the
inter-comparison, the modelled and observed density profiles were interpolated to the same vertical resolution (if the model
resolution is higher than the core resolution, the model is interpolated to fit the core resolution and vice versa). In the SumUp
dataset 96 profiles had the exact date given, 7 SumUp profiles only had year and month given, here the modelled mean density
of the given month were compared, finally 36 cores had only the year given, in these cases the modelled mean density of
January were compared, as we assume they were most likely collected in the middle of the standard Antarctic summer field
season. To evaluate the model performance we calculate Mean Difference (MD) and Standard Deviation (SD) between the
modelled and observed firn densities.

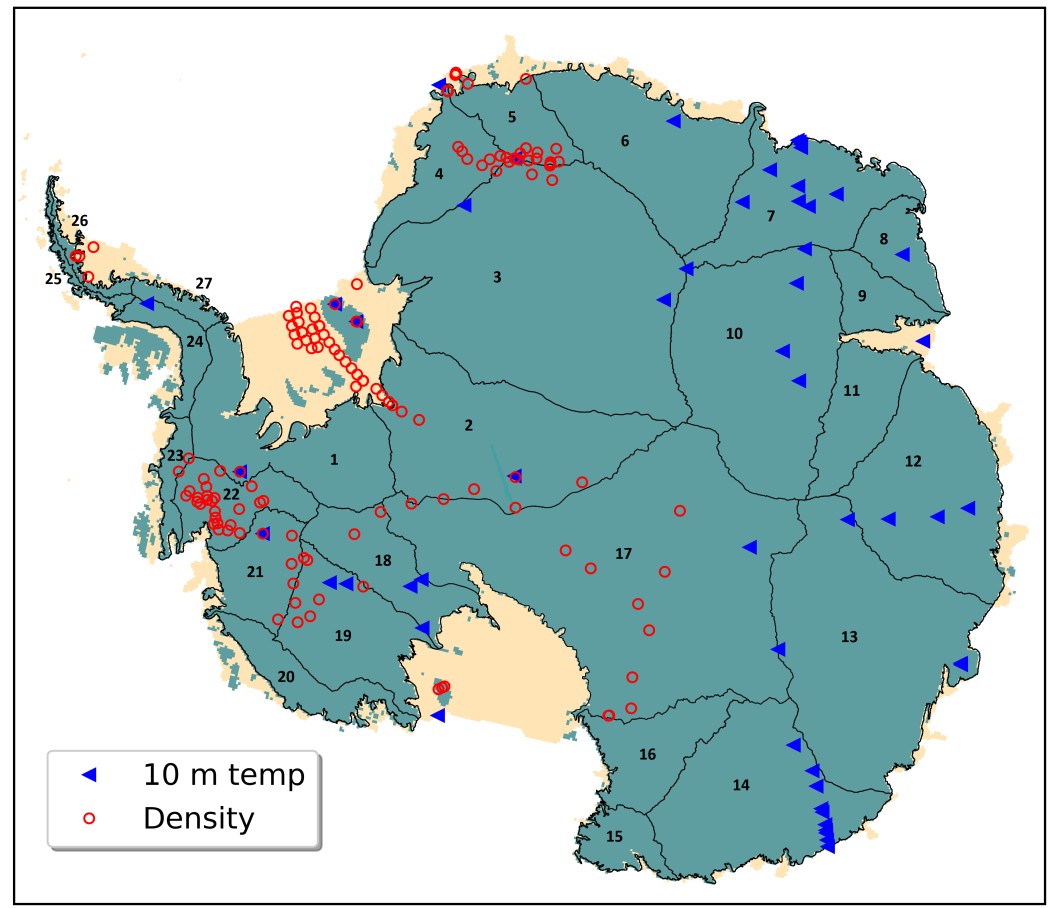

**Figure 3.** The blue-gray colour shows the GAIS and the light brown colours show the locations of ice shelves. The Spatial distribution of observations are shown with blue triangles for borehole temperatures and red circles for the location of the density profiles. The grounded basins are derived from Zwally et al. (2012) and outlined by black lines.

Statistical comparison of mean difference and one standard deviation between the firn cores and the modelled densities, for the three simulations are given in Table 3. Summed up over the AIS all simulations underestimate the densities. However, it is seen that the Fixed version is outperformed by Dyn03 and Dyn15 for densities below than 550 kg m$^{-3}$. For densities higher than 550 kg m$^{-3}$ all three model versions also underestimate the density, with Dyn15 having the largest bias to the observations. The agreement with the in-situ cores also vary spatially (Fig. 4). However generally the density biases are consistent between

the models.

Over the Filchner–Ronne ice shelf, in Dronning Maud Land and in Marie Byrd Land the distribution of profiles are quite dense, these areas are marked with boxes (Fig. 4). All simulations overestimate the density of firn over the Filchner–Ronne ice shelf. Of the 36 cores on the Filchner–Ronne ice shelf, four cores have underestimated densities in the Fixed and Dyn15 simulation, while the rest of the cores have overestimated densities from 2.5 and up to 50 kg m$^{-3}$. Dyn03 has three cores where





**Table 3.** Mean difference between the modelled and observed firn densities (model - core) and standard deviation of the modelled densities above and below 550 kg m$^{-3}$. In total 139 cores were used, see Fig. 3 for locations.

|  | Fixed | Dyn03 | Dyn15 |
|---|---|---|---|
| MD ($\rho <$550 kg m$^{-3}$) | -24.0 | -8.2 | -10.0 |
| SD($\rho <$550 kg m$^{-3}$) | 18.4 | 15.3 | 16.4 |
| MD ($\rho >$550 kg m$^{-3}$) | -31.7 | -32.5 | -35.0 |
| SD ($\rho >$550 kg m$^{-3}$) | 23.4 | 23.1 | 23.7 |

densities are underestimated. Mottram et al. (2020) show that the HIRHAM5 model estimates higher precipitation over the Filchner-Ronne ice shelf than other RCMs, and the overestimate in density may therefore relate to overestimated precipitation in this area. However, as they also note, the lack of SMB observations makes it difficult to be certain if and by how much precipitation is overestimated in this region. In Dronning Maud Land there are 30 cores with a very small bias, the majority of the core densities agree within $\pm25$ kg m$^{-3}$, apart from one core that is underestimated by 100 kg m$^{-3}$ in all three simulations.

Marie Byrd Land shows a general pattern of underestimated densities in 37 cores in all simulations. However, Dyn03 and Dyn15 have lower biases compared to the Fixed. In the Dyn03 and Dyn15 five cores were underestimated more than 100 kg m$^{-3}$, compared to 13 cores in the Fixed model. Both Dyn03 and Dyn15 have three cores where the mean deviations are between 0 and -2.5 kg m$^{-3}$ for densities less than 550 kg m$^{-3}$, but underestimate densities greater than 550 kg m$^{-3}$ with mean deviation between 25 and 50 kg m$^{-3}$.

For the Ross ice shelf cores and near the South Pole, the Fixed simulation underestimates all the cores, and many of them by 50 to 100 kg m$^{-3}$ for densities less than 550 kg m$^{-3}$, and more than 100 kg m$^{-3}$ for densities greater than 550 kg m$^{-3}$. However, for Dyn03 and Dyn15 we also observe an underestimation of most cores, but only a seven them are underestimated by more than 25 kg m$^{-3}$.



**Figure 4.** The density bias between simulations and the observations (model minus core) the outer ring represent densities less than 550 kg m⁻³ and the inner circle represents densities greater than 550 kg m⁻³. Panel (a) is the Fixed model, (b) is Dyn03 and (c) is the Dyn15. Each panel show the entire AIS with three dashed black boxes, each box outlines a zoom-in area, from east to west the Dronning Maud Land, Filchner–Ronne ice shelf and Marie Byrd Land. All displays have the same colourbar.

Figure 5 shows four of the 139 firn cores: core BER02C90_02 (Wagenbach et al., 1994b) (Fig. 5a), core DML03C98_09 (Oerter et al., 2000a) (Fig. 5b), core FRI14C90_336 (Graf and Oerter, 2006h) (Fig. 5c) and core Site 11 (Morris et al., 2017) (Fig. 5d). The simulations fit quite well (±35 kg m⁻³) with the core taken on Berkner Island (Fig. 5a), however the surface density in the model is 125 kg m⁻³ too high. The core from Dronning Maud Land (Fig. 5b) has a high vertical resolution, down



around 4 meters of depth, the Dyn15 has a bias of -2 kg m$^{-3}$, whereas the Dyn03 has a bias of 45 kg $^{-3}$. Cores FRI14C90_336 and Site 11 are taken on the Ronne ice shelf and in Marie Byrd Land respectively. The model densities in FRI14C90_336 are

overestimated below one meter depth, the mean bias is 55 kg m$^{-3}$. At Site 11 all simulations underestimate the density, however below two meters depth, the underestimation is nearly constant with a mean bias of -40 kg m$^{-3}$.

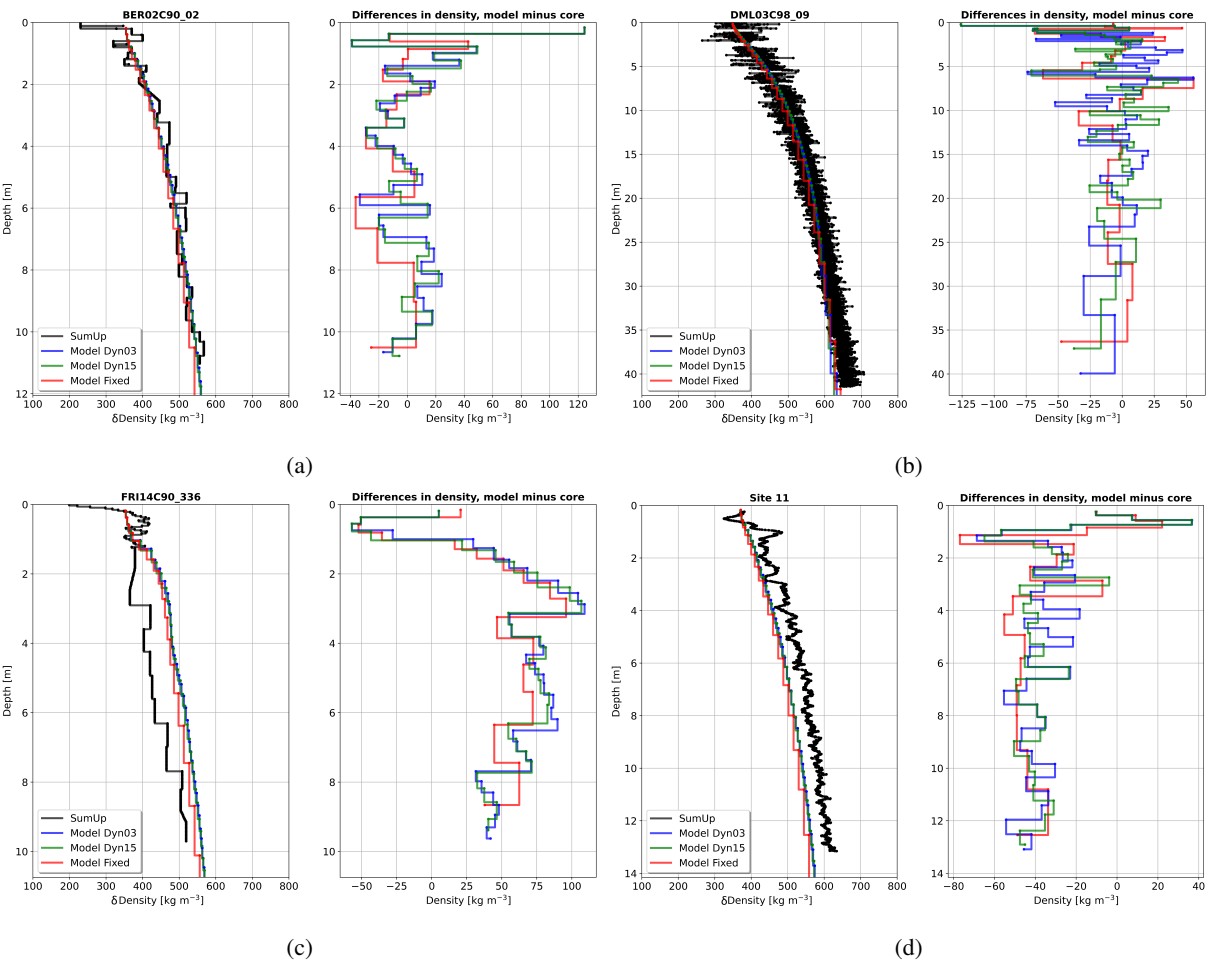

**Figure 5.** Examples of density profiles. In each of the four subfigures, the left hand plot shows the firn core in black and the modelled density from the Fixed, Dyn03 and Dyn15 in red, blue and green. The right hand plot shows the difference. The cores are: 5a BER02C90_02 taken in 1990 (Wagenbach et al., 1994b), 5b DML03C98_09 taken in 1998 (Oerter et al., 2000a), 5c FRI14C90_336 taken in 1990 (Graf and Oerter, 2006h) and 5d Site 11 taken in 2013 (Morris et al., 2017).

The modelled subsurface temperatures are evaluated against observed 10 metre firn temperature measurements from 49 boreholes (van den Broeke, 2008) (see Fig. 3 for the locations). Most of the temperatures were taken in the 1980's and 1990's, however only the year or decade is known for when these were taken. Therefore they are compared with the modelled mean 10

m firn temperature from 1980-2000. We evaluated the model performance using the Root Mean Square Difference (RMSD),



**Table 4.** Mean deviation, root mean square deviation and coefficient of determination, for the modelled and observed 10 meter temperature.

|  | Fixed | Dyn03 | Dyn15 |
|---|---|---|---|
| MD [°C] | 0.42 | 0.52 | 0.46 |
| RMSD [°C] | 1.66 | 1.77 | 1.71 |
| $R^2$ | 0.98 | 0.97 | 0.98 |

Mean Difference (MD) and coefficient of determination ($R^2$). Subsurface temperatures are only sparsely available in Antarctica. The measured 10 m firn temperatures are compared with the modelled mean 10 m firn temperature of the nearest grid cell (Fig. 6). The red, blue and yellow lines are the regression lines of 1$^{st}$ order, for the Fixed, Dyn03 and Dyn15; they have an $R^2$ of 0.98, 0.97, 0.98, respectively. It is assumed that the in-situ temperatures are true, so the errors are in the modelled temperatures. For temperatures below -30 C°the three simulations are in agreement, but in warmer firn temperatures >-30C°, the agreement becomes smaller. The mean deviation of the three model simulations are listed Table 4.

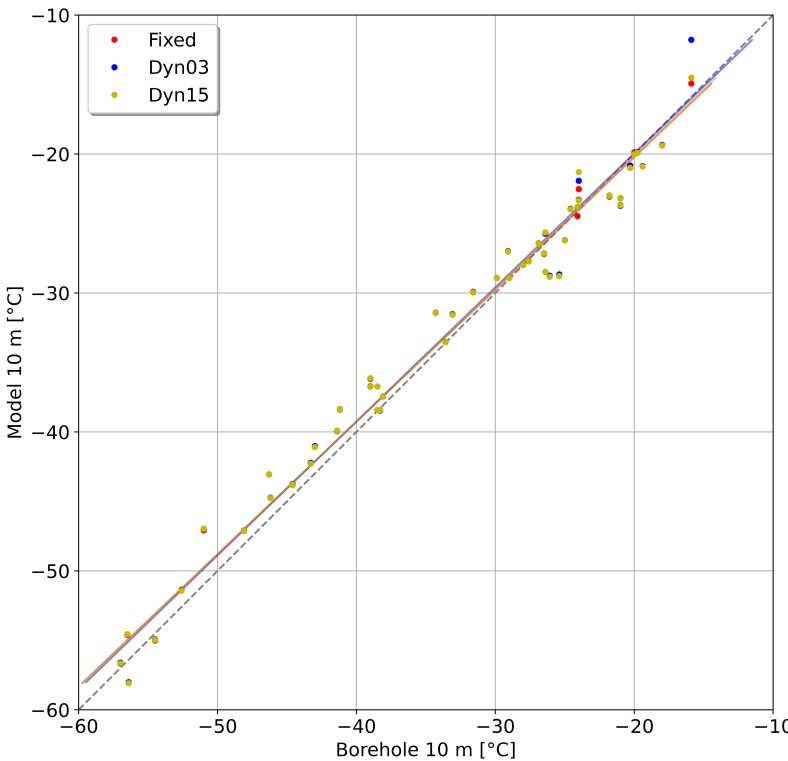

**Figure 6.** The dots are 10 m temperature from boreholes vs. mean model 10 meter temperature. The solid lines are the regression lines of 1$^{st}$ order and the grey dashed line shows the diagonal.



## 4 Discussion

The annual SMB for the three simulations (Table 2) are of the same magnitude as the previous HIRHAM5 SMB estimate of 2659 Gt per year for the ToAIS (Mottram et al., 2020). However, we model a lower SMB, with only Fixed and Dyn03
within one standard deviation range of Mottram et al. (2020). The lower SMB estimates are due to the inclusion of the runoff component in the SMB calculation. The initial SMB result from HIRHAM5 in Mottram et al. (2020) were only calculated from precipitation, evaporation and sublimation. Calculating the SMB by including a subsurface model results in a more realistic SMB, due to the fact that it takes surface and subsurface processes like energy fluxes, meltwater percolation and refreezing into account.

The spatial distribution of SMB broadly agrees with other studies (Van de Berg et al., 2005; Krinner et al., 2007; Agosta et al., 2019; Souverijns et al., 2019). However, the total integrated mean SMB in these published studies differ, likely due to a number of different reasons. The ice mask, model resolution and domain as well as nudging (if any) are identified as a source of differences in Mottram et al. (2020). However, differences in model parameterisations affecting components such as sublimation and precipitation are also important. For example, the modelled annual mean precipitation in HIRHAM5 is 2971
$\pm$ 122 Gt, in COSMO-CLM$^2$ it is 2469 $\pm$78 (Souverijns et al., 2019) and RACMO2.3p2 2396 $\pm$110 Gt (Van Wessem et al., 2018). However, the geographical distribution of precipitation is uneven between these models, with COSMO-CLM$^2$ much drier in western Antarctica than other models in the comparison. Even using a common ice mask, Mottram et al. (2020) found that the difference in precipitation is around 500 Gt per year between HIRHAM5 (the wettest model) and COSMO-CLM$^2$ (the driest model in the intercomparison). The high precipitation in regions of high relief in HIRHAM5 is attributed to a wet
bias in the precipitation scheme, also identified in southern Greenland and similarly occurring in the RACMO2.3p2 regional climate model (Hermann et al., 2018). In both models this wet bias in steep topography is related to the precipitation and cloud micro-physics schemes (Mottram et al., 2020). Areas with a negative SMB can be due to large melt rates, which is what we see in the model over the Larsen ice shelf with melt values between -1200 and increasing toward west to -2300 mm w.eq. per year and SMB values in the range of -300 to -1800 mm w.eq. per year increasing toward west. Negative SMB values can also
be due to high sublimation rates in e.g., blue ice areas (Hui et al., 2014). For exampe,Kingslake et al. (2017) found blue ice in Dronning Maud Land and near the Transantarctic Mountains, in these areas our SMB model mean also shows negative SMB between -50 and -400 mm w.eq. per year. A closer investigation (not shown) reveals that the negative SMB values in these areas are driven by the sublimation and thereby consistent with the creation of blue ice areas.

The differences in SMB between the model simulations (Fig. 1b) are largest near the coast in West AIS and especially on
the Larsen ice shelf. This is confirmed in Fig. 2b, where the difference in integrated SMB between the model simulations are greater when the ice shelves are included. We attribute the differences between the Fixed and Dyn models to the following differences in model designs: The increased vertical resolution in the Dyn models, with a higher vertical resolution (the top layers can be 6.5 centimeter w.eq. thick) means that the cold content in the upper layers is depleted faster and it starts to melt while the layer below is potentially still below freezing. Whereas the top layers in the Fixed model get thicker rather quickly,
which means it takes longer to be brought to melting point and it starts melting. Furthermore, the two versions of the subsurface



model, have different melting schemes. The Fixed model melts snow first and then, if there is more energy left, melts the ice. Whereas, the Dyn melts snow and ice simultaneously. This simultaneous melting of snow and ice was introduced in the Dyn version to prevent the top layer being depleted of its snow content and left only with ice (Vandecrux et al., 2018). A top layer composed of ice would then prevent surface melt to infiltrate below the top layer. By melting snow and ice simultaneously,

there is always snow in the top layer for meltwater infiltration to happen. This difference of infiltration may cause the snow-melt first to refreeze less and runoff more water than the snow-and-ice-simultaneous melt. To investigate these differences in melt, refreezing and runoff, the runoff fractions have been plotted in figure 2f and listed in table 2. Here it is seen that even though the difference in melt between Dyn03 and Dyn15 is only around 20 Gt per year, the difference between the runoff to melt fractions are larger. The Fixed model melts around 300 Gt per year less than the dynamical versions but the runoff to

melt ratio is the same for the Fixed and Dyn15. This means that the Fixed and Dyn15 have the same relative runoff, leading to the same relative refreezing, indicating that this difference does not cause significant partition of melt between refreezing and runoff.

The difference in SMB between the three simulations confirms how complex it is to estimate the SMB. Just by changing the subsurface scheme the final result differs by 90 Gt per year. By keeping the same subsurface scheme and changing the spin-up

length the final result differs by 110 Gt per year. These changes in SMB illustrate the consequences of including dynamical firn processes since the layer density and temperature and other firn properties are better conserved, potentially allowing more retention and refreezing where there is capacity or reducing it where there is not. Although these differences are currently only a few percent of the total SMB, as the climate warms and melt becomes more widespread in Antarctica e.g. (Boberg et al., 2020; Kittel et al., 2020), accounting for these processes will become more important. Moreover, on a local and regional scale,

the differences are more important when determining mass balance in basins or outlet glacier/ice shelves.
The differences between versions with a different spin-up period suggest that the snowpack is not quite in equilibrium in all locations. Therefore, SMB calculations consequently vary due to the amount of melt calculated during the initialisation period. Retention and refreezing of meltwater during spin-up causes different profiles of temperature and density to develop depending on how long the spin-up lasts for. These results therefore emphasise the importance of adequate spin-up and assessment of the

effects of snowpack spin-up in producing and using SMB in Antarctica.

Vandecrux et al. (2020b) found that the Fixed version smoothes the firn density profiles, when compared to the dynamical version, this is confirmed by our results. One of the criteria for the dynamical version is that it prefers to merge layers deeper than 5 m of w.eq., meaning that the top 5 m w.eq. have a high vertical resolution, this makes it easier to detect changes in density. In areas such as the AP, Ronne-Filchner ice shelf, Ross ice shelf and in coastal areas of Dronning Maud Land where

seasonal melt occurs (Zwally and Fiegles, 1994; Wille et al., 2019), meltwater can percolate into the firn and refreeze, creating ice lenses that change the density, but that cannot be detected if the subsurface scheme have layers with a fixed mass even if the vertical resolution is increased (Vandecrux et al., 2020b). Not only is there a difference between the models when evaluating density profiles, this study also shows the importance of spatial evaluation, here the three simulations follow the same pattern by over/underestimating the densities in the same areas (Fig. 4). This systematic bias may indicate either further tuning of

densification routines are necessary or that there are systematic biases in accumulation, leading to these errors. The subsurface





scheme does not currently incorporate wind-blowing snow processes that may prove important also in correcting biases in accumulation. On the other hand, although 0.11 degree is a high resolution model in Antarctica and thus better captures topographic variability than lower resolution models, it is still relatively coarse when it comes to capturing steep topography. Errors in orographic precipitation are difficult to measure even in well instrumented basins and are poorly captured in Antarctica
where observations are few and far between. The densification bias becomes especially important when using altimetry data to estimate the total MB, like in Shepherd et al. (2018) and Rignot et al. (2019), here the firn densification rate is needed to correct the altimetry data (Griggs and Bamber, 2011).

Since the density cores are primarily taken from West Antarctica and Dronning Maud land, these statistics represents complex areas with high precipitation and melt/refreezing events, whereas, density comparisons from less complex areas (low
precipitation and no melt/refreezing) such as East Antarctica are sparse. Nonetheless they are still very important. Based on the statistics from these model set ups, the Dyn version is preferred when modelling densities below 550 kg m$^{-3}$, at higher densities the differences between the model versions become very small.

The simulated 10 m firn temperature depends on the thickness and numbers of the layers above the 10 m point. The thickness of a layer determine how conductive heat fluxes are resolved in the near-surface snow. A thicker layer will have more thermal
inertia and will require more energy to be warmed up. A thin layer can respond much quicker to fluctuations in the surface energy balance. Differences in simulated temperatures between models, as we see in Table 4 can therefore be explained by vertical resolution, which affects both their calculation of temperature and how the heat is conducted to a depth of 10 m. Note that the models also use different thermal conductivity parameterisations.

## 4.1 Satellite gravimetric mass balance

Over the AP there is a large disagreement between SMB-D$_{Gardner}$ and SMB-D$_{Rignot}$, the mean discharge values differ 90 Gt per year, D$_{Rignot}$ being the largest, this results in opposite trends of SMB-D. SMB-D$_{Gardner}$ shows a mass gain of around 600 Gt and SMB-D$_{Rignot}$ shows approximate mass loss of 1150 Gt over the period, whereas GRACE has a mass loss of around 400 Gt for the period (Fig. 7A). There are times where the variability between GRACE and the two SMB-D graphs follow each other, e.g local peak around year 2006, 2011 and 2017. Since the discharge is plotted as a constant, this variability originates
from the SMB model, most likely precipitation, this means that either D$_{Gardner}$ value is too small, D$_{Rignot}$ values are too large or that the SMB magnitude is too low/high depending on which discharge is used. As the resolution of GRACE is quite coarse, it can add to the uncertainties over the AP, because of narrow topography. Over the grounded West AIS the trend of GRACE, SMB-D$_{Rignot}$ and SMB-D$_{Gardner}$ agrees, they all see a mass loss, of around 2000 Gt, 2150 Gt and 1700 Gt, respectively, for the overlapping period (Fig. 7B). The discharge values from the two studies differ only 2 Gt per year from 2002 to 2010, but
50 Gt per year from 2010 to 2017, with D$_{Gardner}$ being the lowest. GRACE measures a smaller mass loss in the beginning of the period, then around 2009 the GRACE mass loss increases. Both Gardner et al. (2018) and Rignot et al. (2019) have found an increasing discharge in West Antarctica. However due to the limited temporal resolution from Gardner et al. (2018) the discharge is assumed constant, resulting in a equal offset in SMB-D from 2002-2009, but then diverging results from 2010.





This show that in areas where there are large changes in the dynamical mass loss, discharge values with a higher temporal
resolution are needed.

Over the East GAIS the agreement between GRACE and SMB-D$_{Gardner}$ is remarkably good. Between 2009 and 2011 large
snowfall events were observed in Dronning Maud Land (Boening et al., 2012; Lenaerts et al., 2013) (basins 5-8 in Fig. 3).
These snowfall events lead to rapid mass gain, which is seen in both GRACE and SMB-D$_{Gardner}$, especially in 2009-2010
(Fig. 7C). This mass gain is less pronounced in SMB-D$_{Rignot}$ because it estimates an overall mass loss for the period. In the
SMB signal there are yearly variabilities, however, these variabilities are larger in the GRACE data compared to SMB-D. For
the entire GAIS GRACE is detecting a mass loss of 900 Gt, SMB-D$_{Gardner}$ shows a mass gain of 500 Gt and SMB-D$_{Gardner}$
shows a mass loss of 4000 Gt, for the overlapping period 2002-2017. The majority of that difference between GRACE and
SMB-D$_{Gardner}$ can be attributed to the AP, the difference between GRACE and SMB-D$_{Rignot}$ arise from the AP and East GAIS.

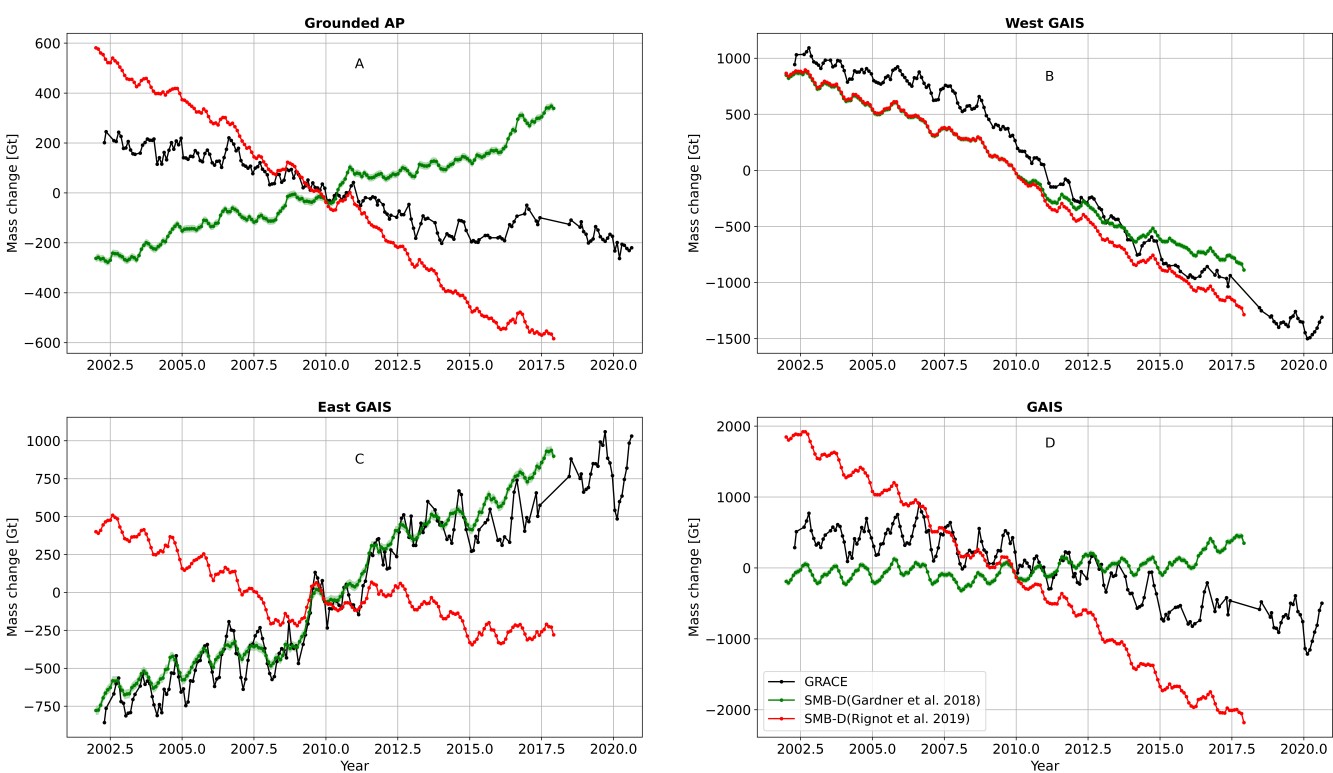

**Figure 7.** Integrated relative mass change over the grounded Antarctic Peninsula (A), the grounded West AIS (B), grounded East AIS and
the GAIS (D). GRACE relative mass change from 2002 to 2020 (black graph). SMB minus discharge (green/red graphs), SMB values are
from the Dyn15 simulation, discharge values are derived by Gardner et al. (2018) (in green) and Rignot et al. (2019) (in red). Note that the
y-axis differ from panel to panel.



## 4.2 Circulation effects on SMB and the Southern Annular Mode

We observe a robust (outside the 5-95[th] percentile range) relationship between SMB and SAM in 13 out of the 27 basins (Fig. 8a,b). For each phase of the SAM, SMB anomalies (SMBa) are defined as the SMB in months with a SAM-or SAM+ monthly mean minus SMB over the full period. The SMBa for SAM-, when the westerlies are further away from the Antarctic continent, show magnitudes well outside the percentiles for all model simulations. Basins 2 and 3 that have outlets to the Filchner–Ronne ice shelf (EAIS), basin 4 in Dronning Maud Land (EAIS), basin 7 in Enderby land (EAIS), basins 9, 10, 12 surrounding the

Amery ice shelf (EAIS), basins 17, 18, 19 with outlet into Ross ice shelf, 20 in Marie Byrd Land (WAIS) and basins 24, 25 located on the windward side of the AP are particularly affected by SAM-. For SAM+, when the westerlies are closer to the continent the SMBa magnitudes are generally smaller and have an opposite sign, however we see the same pattern in the same basins as for SAM-. A SAM+ phase results in a relatively low pressure over the AIS compared to the mid latitudes, and we see a negative SMBa in 16 of 27 basins, namely: 6, 9-13, 15-23 and 26 (Fig. 8b). Marshall et al. (2017) reported a similar

signal for precipitation, which confirm our results since precipitation is the main driver of SMB. Basin 26 shows a negative SMBa and basin 27 has a slightly positive SMBa, this is due to the steep orography on the windward side of the AP creating a shadowing effect on the leeward side of the AP. For SAM- the SMBa signal is opposite and the mean magnitude of the signal is 26% larger in all basins (Fig. 8a). During months of SAM+ the average SAM index is 1.45. Figure 8 show that basins 1-5, 7, 14, 24-25 and 27 have SMB anomalies (SMBa) of the same sign as the SAM: SMB is 0.28 Gt per year higher than average

in case of SAM+ and 0.39 Gt per year lower than average in case of SAM-. Those basins are mostly located in the East, Ross and Amundsen sea sectors. Contrastingly, basins 6, 8-13, 15-23 and 26 have SMB 0.32 Gt per year below average in months of SAM+ and 0.43 Gt per year above average in months of SAM-. These basins are mostly located in the Weddell sea, Dronning Maud Land and Bellinghausen sectors. For months with SAM- the average SAM index is -1.36. We can see that for SAM of similar absolute magnitude, SAM- has a stronger impact on SMB over the GAIS.

In both positive and negative SAM events basins 24 and 25 on the windward side of the AP, show strong correlation between the SAM index and SMB magnitude also reported by Marshall et al. (2017). So even though the AP is narrow (50 to 300 km across) the SAM plays an important role. From 1980 to 2017 the SAM has become more positive (Fogt and Marshall, 2020), this positive trend in the SAM is attributed to stratospheric ozone depletion (Thompson et al., 2011; Fogt et al., 2017). If this trend continues the basins on the leeward side of the AP will see a smaller mass gain in the future, which could accelerate

the collapse of the Larsen ice shelf. This is also seen in basin 9, 10, 11 and 12 surrounding the Amery ice shelf and basin 21 where Thwaites glacier is located. Not all of the above mentioned basins show $\delta$SMB signals that are statistically robust (i.e. the signals are within the 5[th] or 95[th] percentiles), but if the trend in the positive SAM continues, it might become an important factor in the future (Fogt and Marshall, 2020).

It is thus important to take the SAM phases into account when investigating the SMB at regional scale, furthermore it is

extremely important that global circulation models resolve the SAM realistically if future climate projections are to be used with confidence to make projections of sea level rise from Antarctica.





(a)

(b)

**Figure 8.** SMBa (monthly values minus mean values) in months for SAM- (a) and SAM+ (b), for each basin. The vertical dashed lines split the basins into areas. Starting from left, we show basins towards the Weddel sea, Dronning Maud Land, Eastern coast, Ross sea, Western coast/Admundsen sea and Bellinghausen sea. The thin bars are the 5[th] and 95[th] percentiles, for the bootstrapping analysis with 1200 runs. Locations of the basins can be seen in Fig. 3.



# 5  Conclusions

We estimate the Antarctic SMB ranging from 2583.4±121.6 and 2473.5±114.4 Gt per year over the total area of the ice sheet including shelves and between 1995.4±99.3 and 1963.3±96.2 over the grounded part, for the period 1980 to 2017. The difference is due to different subsurface models forced with HIRHAM5 outputs. The Dyn03 version has the highest integrated SMB over the ToAIS (GAIS) 2583.4±121.6 (1995.4±99.3) Gt per year, Dyn15 has the lowest 2473.5±114.4 (1963.3±96.2) Gt per year. The Fixed version is ≈19 Gt per year lower than Dyn03 over the ToAIS and 0.2 Gt per year lower than Dyn03 over the GAIS. The simulations compute nearly equal SMB over the interior, the main differences are seen in the coastal areas of West AIS and the AP. The Dyn15 simulation gives the smallest SMB estimate and is thus closest to other studies (Van Wessem et al., 2018; Souverijns et al., 2019; Agosta et al., 2019), however it is still 200-300 Gt per year higher. Evaluating the modelled density profiles show the Lagrangian model set up has the lowest bias and standard deviation in density differences for densities less than 550 kg m$^{-3}$, but for densities above 550 kg m$^{-3}$ the performance of the models are nearly identical. In general all models overestimate the densities on the Flicher-Ronne ice shelf and underestimate the densities in Marie Byrd Land and around the Ross ice shelf. It is therefore clear that there are regional systematic biases. To evaluate our simulated SMB, we compare our simulations to MB estimations (SMB minus discharge) from GRACE. We use discharge from two sources: Gardner et al. (2018) and Rignot et al. (2019). There are large differences between the discharge values over the AP leading to our simulations overestimating MB when using $D_{Gardner}$ and underestimating MB when using $D_{Rignot}$. Over the East GAIS the MB is underestimated using $D_{Rignot}$, but fit quite well to the GRACE MB when using $D_{Gardner}$. These disagreements between the two observational data sets makes it hard to distinguish how well the modelled SMB fits with total mass balance estimates.

Regional precipitation is strongly linked to the phase of the SAM as shown by the bootstrap analysis. By using outputs from HIRHAM5 forced with ERA-interim to resolve the SAM correctly, robust signals are identified in 13 out of 27 basins. It is clear that the phase of the SAM affects the spatial distribution of SMB. When SAM is negative, there is a lower SMB on the windward side of the Antarctica Peninsula and a higher SMB over the plateau and vice versa when SAM is positive. This makes the SAM an important factor to evaluate in global models when downscaling models for projecting future Antarctic climate.

*Code availability.* A Matlab version of the subsurface model code used in these study is available here: https://doi.org/10.5281/zenodo.4178985 (Vandecrux, 2021)

*Data availability.* A selection of the HIRHAM5 data is available here: http://ensemblesrt3.dmi.dk/data/prudence/temp/RUM/HIRHAM/ANTARCTICA/ and a selection of the subsurface model data is availible here: http://ensemblesrt3.dmi.dk/data/prudence/temp/nichan/SMB_paper_data/ during the open discussion, a permanent link will be provided upon acceptance of the manuscript.



*Author contributions.* NH, SBS and RM conceived the study. NH ran the subsurface model simulations with help from PL, made the plots, performed the analysis under the guidance from RM, SBS, PL, PT, BV and RF. HIRHAM5 model simulation was carried out by FB and RM. The Fixed model was developed by PL and the dynamical version were developed by BV and PL. RF prepared the GRACE data. All
450 authors contributed to the manuscript.

*Competing interests.* We declare no competing interests.

*Acknowledgements.* The present work is a contribution to the EU Horizon 2020 PROTECT project, contribution number XX. Data analysis was supported by the Danish State through the National Centre for Climate Research (NCKF). HIRHAM5 regional climate model simulations
were carried out by RM and FB as part of the ice2ice project, a European Research Council project under the European Community's Seventh Framework Programme (FP7/ 2007-2013)/ ERC grant agreement 610055. GRACE data analysis was supported by ESA Climate change initiative for the Greenland ice sheet funded via ESA-ESRIN contract number 4000104815/11/I-NB and the Sea Level Budget Closure CCI Project funded via ESA-ESRIN contract number 4000119910/17/I-NB. We also gratefully acknowledge the ESA CCI Ice sheets project as a forum for the interchange of ideas that led directly to this study.



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
