# Peer review of "Downscaled surface mass balance in Antarctica: impacts of subsurface processes and large-scale atmospheric circulation"

_The Cryosphere, 2021_

## Referee Comment (RC2)

**Downscaled surface mass balance in Antarctica: impacts of subsurface processes and large-scale atmospheric circulation by Hansen et al., 2021**

Hansen et al. reconstructed the Antarctic surface mass balance using two versions of an offline subsurface model forced by HIRHAM5. These two versions have a common basis but differ in the way the snow layers are discretized and in the way melt is taken into account. They found SMB estimations similar for both subsurface models and is comparable to estimates from RCMs. Model results are compared to observed firn densities while SMB is evaluated against GRACE using two ice discharge estimations. Finally, the relation between SAM and SMB is investigated.

The differences between the two versions of the subsurface model are clearly presented, especially the vertical discretization of each version, as well as HIRHAM5 inputs, comparison data and the experimental set-up. Each reconstruction using the two versions (and spinup variation) leads to a coherent SMB even if some components are outside of the range suggested by previous estimations. The relation with SAM and SMB not only confirmed other studies but add more details notably in the relation of each basin with this atmospheric index. This is an interesting paper paving the way to SMB estimates relying on (sub)surface models with a higher resolution. However some comments should be addressed before publication.

**Major comments**

Since the subsurface model is forced by HIRHAM5, how does the use of HIRHAM5 affect the results? The subsurface model is forced by water (evaporation, sublimation) and energy (latent and sensible heat, downwelling longwave) fluxes that are strongly linked to the surface state as computed by HIRHAM5 (and in general for the method, by a model with a less sophisticated surface scheme than the one from the subsurface model). It looks like a vicious circle where results from the subsurface model might be not independent from the surface scheme of the forcing model.

The evaluation of the physical conditions (densities and temperature) of the snowpack is exhaustive however I wonder if the main product of the paper (the SMB) is sufficiently evaluated. Considering all uncertainties in GRACE and discharge estimations, the present evaluation should rather be a supplement comparison than the main part of the evaluation. I recommend the authors to evaluate their SMB against in-situ local observations. It might also help to assess the added value of using the subsurface model instead of the HIRHAM5 SMB by comparing their reconstruction over the observation (see first comment).

Although SAM influences on SMB is still an open question, other studies (ignored in this manuscript) have attempted to contribute to answer this question. I suggest the authors to add some references in their introduction and discussion to better situate their work in the existing literature. Here is a list of some potentially interesting references on the subject (all may not have the same relevant level and probably do not need to be included, and the list is far for being exhaustive):

Dalaiden, Q., Goosse, H., Lenaerts, J. T., Cavitte, M. G., & Henderson, N. (2020). Future Antarctic snow accumulation trend is dominated by atmospheric synoptic-scale events. Communications Earth & Environment, 1(1), 1-9.

Kim, B. H., Seo, K. W., Eom, J., Chen, J., & Wilson, C. R. (2020). Antarctic ice mass variations from 1979 to 2017 driven by anomalous precipitation accumulation. Scientific reports, 10(1), 1-9.

Medley, B., & Thomas, E. R. (2019). Increased snowfall over the Antarctic Ice Sheet mitigated twentieth-century sea-level rise. Nature Climate Change, 9(1), 34-39.

Previdi, M., & Polvani, L. M. (2017). Impact of the Montreal Protocol on Antarctic surface mass balance and implications for global sea level rise. Journal of Climate, 30(18), 7247-7253.

Thomas, E. R., Van Wessem, J. M., Roberts, J., Isaksson, E., Schlosser, E., Fudge, T. J., ... & Ekaykin, A. A. (2017). Regional Antarctic snow accumulation over the past 1000 years. Climate of the Past, 13(11), 1491-1513.

Vannitsem, S., Dalaiden, Q., & Goosse, H. (2019). Testing for dynamical dependence: Application to the surface mass balance over Antarctica. Geophysical Research Letters, 46(21), 12125-12135.

**Minor comments**

**P10 L216-217**: This should be verified by comparing the albedos of the different simulations. How is the albedo prescribed in subsurface models? Is it the same parameterisation as in HIRHAM5? There is no information on this subject whereas the albedo is a determining parameter and will be even more so in a warmer climate.

All the experiments reveal particularly high melt values that are significantly different from other estimations based on RCMs (eg., Van Wessem et al., 2018; Agosta et al., 2019; Kittel et al., 2021) or satellites (eg., Trusel et al. 2013). It does not mean that these values are erroneous since there are by definition no observations of melting, but they deserve further discussion even if they have no impact on the SMB in the current climate.These large differences in the present climate might suggest that the model cannot be used in a warmer climate where melting and runoff would have much more impact. The authors could compare their estimates with SEB model estimates forced by AWS (Jakobs et al., 2020) or any other estimates.

**Specific comments**

Gt per year: to be consistent with kg m$^{-3}$, consider Gt yr$^{-1}$
**P1 L11-L14**: consider to remove the section about the density and temperature biases from the abstract as it does not seem to be a particularly important information.

**P2 L32**: Add blowing snow erosion/deposition in the SMB definition or specify that is naturally included in the local¨ solid precipitation balance.

**P2 L40-43**: RCMs also improve the physical representations of specific processes over polar areas (see for instance Lenaerts et al., 2019).

**P2 L43-44**: "Mottram et al. (2020) evaluated the atmospheric output from five different RCM simulations of Antarctic SMB driven by ERA-Interim (1987-2017)."
Atmospheric output vs SMB (= surface output) is confusing, please rewrite.

**P2 L44-47**: Indicating the original values of the models does not provide much more information since these are the models that were used in Mottram et al., 2020. It is more of a repetition with perhaps less relevant information because the masks are different (i.e. the SMB is also different, whereas this artifact is corrected in Mottram et al., 2020). I would remove the individual values, and if the authors still want to link this comparison study with original model publications, they could cite the name of the models (+reference) that were used in Mottram after the SMB ranges.

**P3 L65-67**: Please add some references here.

**P3 L66-70**: Even is SAM has indeed a strong effect on precipitation patterns, Marshall et al. (2017) rather suggest that precipitation patterns result from a combination of the different modes. Consider add other references to better justify the selection (ie, Kim et al., 2020)

**P4 L100-101**: "Despite the forcing is based on 6 hourly values the subsurface scheme is used to simulate the subsurface at 1-hour time steps."
Is there an interpolation between two 6-hourly inputs to produce a smooth transition between two forcing time steps?

**P4 L105**: Please specify if this is vertically or also laterally transferred?

**P5 L131**: Replace weighed by weigh**t**ed

**P5 L137**: Could you be more specific? Does this mean that the melting is taking place on several layers in the vertical at the same time, i.e. that the energy is transmitted into the snowpack?

**P5 L141-142**: What climatological means did you use? Is it based on Hirham5 inputs?

**P5 L144**: Why did you initialized with a uniform density over the whole ice sheet, that is close to snow values given by Kasper et al. parameterisation? I guess that spinup time remove the dependency to the initialization but it could have been more consistent?

**P12 L240-242**: Could it also due to overestimated melt/refreezing (see minor comment #2). Overestimated precipitation could also result in more fresh snow with lower/uncompacted densities. It could also result from an overestimation of the "fresh" surface snow density (linked to the parameterisation itself or HIRHAM biases)

**P14 L54**: Why did you select these specific cores? Are they the only ones with a high vertical resolution or are they representative of the region? It would be interesting to state the objectives of this comparison in order to extrapolate the conclusions that can be drawn from these few examples.

**P16 L293**: Melt is not a balance, there are either no melt or melt and then positive values.

**P17 L306**: Do you mean that one layer in the pack can be ice and snow at the same time or that melt occurred at different vertical layers simultaneously?  (see also P5L137)

**P20 L399-404**: Check the SMBa units, shouldn't it be Gt m$^{-1}$ (as monthly SAM values or as Figure 8) instead of Gt yr$^{-1}$ ?

**Figures and Tables**

**P6**- **Table 1**: replace fist by fi**r**st

**P8**- **Figure 1**: The colormap is confusing as it is non-continuous. Please select something with a linear transition that will allow the reader to easily identify SMB variations. (See for instance https://matplotlib.org/stable/tutorials/colors/colormaps.html,  perceptually uniform colormaps or sequential colormaps ). Consider to use the abbreviation you defined for water equivalent (and similar remark than the first specific one).

**P13 – Figure 4**: similar remark than Figure 1 about the colormap + consider to make it bigger (maybe on a whole page with the elements one below the other?) to improve the reading.

**P21 Figure 8**: add the time unit relative to the SMB accumulation (m$^{-1}$)

**Stylistic suggestions** (feel free to refuse them all if you wish without justification)

**P1 L3-4** : I suggest '[ ...] in determining SMB before *influencing* the total mass balance of Antarctica and global sea level *variations.'*

AIS SMB : Maybe the authors could avoid the repetition of a double accronym and could keep the form used at the beginning of their abstract (Antarctic SMB).

**P1 L19-20**: 'Finally, we compare the modelled SMB to GRACE data by subtracting the solid ice discharge. *We* find a good agreement in East Antarctica, but large disagreements over the Antarctic Peninsula *potentially caused by* large difference between published estimates of discharge that make it challenging to use mass reconciliation in evaluating SMB models on the basin scale'.

**P2 L46-57**: Please, try to decrease the occurrence of "show".

**P2 L53**: "full-subsurface"

**P2 L54**: "over 1979-2017'' … « Antarctic SMB »

**P4 L96** : replace Thereby (already used at the previous sentence)

**P4 L108** : replace «layer's conductivity » by layer conductivity or conductivity of the layer (similarly for P5L125/6)

**P6 L172** : Try to avoid repeating the « To see if » structure (two times in this sentence and already a bit earlier)

**References**

Agosta, C., Amory, C., Kittel, C., Orsi, A., Favier, V., Gallée, H., van den Broeke, M. R., Lenaerts, J. T. M., van Wessem, J. M., van de Berg, W. J., and Fettweis, X.: Estimation of the Antarctic surface mass balance using the regional climate model MAR (1979–2015) and identification of dominant processes, The Cryosphere, 13, 281–296, https://doi.org/10.5194/tc-13-281-2019, 2019.

Jakobs, C. L., Reijmer, C. H., Smeets, C. P., Trusel, L. D., Van De Berg, W. J., Van Den Broeke, M. R., and Van Wessem, J. M.: A benchmark dataset of in situ Antarctic surface melt rates and energy balance, Journal of Glaciology, 66, 291–302, 2020.

Kim, B.-H., Seo, K.-W., Eom, J., Chen, J., and Wilson, C. R.: Antarctic ice mass variations from 1979 to 2017 driven by anomalous precipitation accumulation, Scientific reports, 10, 1–9, 2020.

Kittel, C., Amory, C., Agosta, C., Jourdain, N. C., Hofer, S., Delhasse, A., Doutreloup, S., Huot, P.-V., Lang, C., Fichefet, T., and Fettweis, X.: Diverging future surface mass balance between the Antarctic ice shelves and grounded ice sheet, The Cryosphere, 15, 1215–1236, https://doi.org/10.5194/tc-15-1215-2021, 2021.

Lenaerts, J. T., Medley, B., van den Broeke, M. R., and Wouters, B.: Observing and modeling ice sheet surface mass balance, Reviews of Geophysics, 57, 376–420, 2019.

Trusel, L. D., Frey, K. E., Das, S. B., Munneke, P. K., & Van Den Broeke, M. R.. Satellite-based estimates of Antarctic surface meltwater fluxes. Geophysical Research Letters, 40(23), 6148-6153, 2013.

van Wessem, J. M., van de Berg, W. J., Noël, B. P. Y., van Meijgaard, E., Amory, C., Birnbaum, G., Jakobs, C. L., Krüger, K., Lenaerts, J. T. M., Lhermitte, S., Ligtenberg, S. R. M., Medley, B., Reijmer, C. H., van Tricht, K., Trusel, L. D., van Ulft, L. H., Wouters, B., Wuite, J., and van den Broeke, M. R.: Modelling the climate and surface mass balance of polar ice sheets using RACMO2 – Part 2: Antarctica (1979–2016), The Cryosphere, 12, 1479–1498, https://doi.org/10.5194/tc-12-1479-2018, 2018.

---

## Author Comment (AC2)

Reply to reviewer comments on

**"Downscaled surface mass balance in Antarctica: impacts of subsurface processes and large-scale atmospheric circulation"**

by

Nicolaj Hansen, Peter L. Langen, Fredrik Boberg, Rene Forsberg, Sebastian B. Simonsen, Peter Thejll, Baptiste Vandecrux, and Ruth Mottram

Dear Editor Alexander Robinson,

On behalf of my co-authors and myself, I would like to thank the two reviewers for their comments on our manuscript. The reviewers have made an extensive review of the manuscript language and we have followed their suggestions to our best efforts. In the following, we provide a point-by-point answer to all the issues raised by the reviewers. We have gathered and numbered all issues raised by the reviewers (Anonymous Referee #1: 1-29 and Referee #2 Christoph Kittel: 30-56), all issues will be followed by our suggestions for improvement to the manuscript highlighted in red.

We have added a figure in section 3.1, this figure is the new figure 3. This means that the old figure 3 is now figure 4 ect. Furthermore, we discovered a small bug in our firn density calculations so the numbers in table 3 and the text around it have been updated.

Best regards, Nicolaj Hansen

**Anonymous Referee #1**

 First, the manuscript and the reader's ability to understand and assess the data in them, would be improved if some more attention was given to the figures' presentation. Much of this can be summarised as a making a better choice of colour scales and markers. In particular figures 1 and 4 rely on colour scales that do not steadily increase in their darkness, but instead jump around somewhat. This makes assessing more negative and more positive values difficult. Following on from this I would suggest the authors ensure the colours used for the Fixed, Dyn03, and Dyn15 results are consistent across figures (and ideally different from the colours used in the diverging scales). Likewise improvements can be made in the maps in figures 3 and 4: a clearer indication of the zoomed areas, which will be aided by a better choice of background colour and non-biased colour scales. Throughout the figures I recommend that the authors use larger text and labels and better and more consistent labelling of subfigures.

That is a good idea, the colour scales/choices have been changed, as well as the text and labels size. The new figures are shown under the specific comments.

2. Second, and more scientifically interesting, is the question of resolution. Much effort has gone into considering different layer schemes and how mass is transferred between them down to a scale of 0.065 m w eq. In the horizontal direction however the resolution is 12.5km. Clearly there are computational demands that limit this resolution, but it does raise a question: namely, given that some features that indicate or result from localised melt and mass loss can be of this same length scale or less, how is this handled in the model? Is it simple averaging per pixel? Is a higher resolution used in some geographic areas? Or are the smallest of these features simply not seen / modelled? The model has been tuned to mimic the average behavior of the ice sheet surface at 5-12km scale. It cannot resolve subpixel processes. However, the small scale features caused by surface melt (as lakes and streams) translate into an increase of water content in the model.

To clarify this in the manuscript we suggest adding this in line 100 ".... removed from the results. Furthermore, the model has been tuned to mimic the average behavior of the ice sheet surface at a 5-12 km scale. It cannot resolve subpixel processes. However, the small-scale features caused

by surface melt translate into an increase of water content in the model. Despite the forcing....".

3. A final general point: given that the manuscript focuses on firn and surface processes for which surface radiation fluxes are important, the treatment of the albedo seems a little simplistic. Perhaps the authors could expand on this. I understand that computationally it is probably hard to go beyond a broadband albedo, but are the values stated extremes, or are they allocated for each class of surface i.e. all fresh snow is designated as having an albedo of 0.85, or is this dependent on grain size, density etc? Yes, indeed it is simplistic, we are planning to do more work on the albedo scheme later this year. However, it could be described better in the manuscript, this will be added starting in L87: "Following Langen et al. (2015) the shortwave albedo is computed internally and uses a linear ramping of snow albedo between 0.85 below -5 C and 0.65 at 0C for the upper-level temperature. The albedo of bare ice is constant at

0.4. Furthermore, a transition albedo is calculated for thin snow layers on ice, based on Oerlemans and Knap (1998) with an e-folding depth of 3.2 cm for snow."

4. I think adding some comments on both the horizontal resolution point and some further details of how albedo is treated in the model would assist the reader with a general glaciology background, but who is less-versed in the details of such models.

Yes, that is a good idea, we have done that, see answers to comment 2 and 3

- I29 "as such might" -> "due to their role in" Changed
- I57 "Acknowledging that the in-situ observations might be challenged judging the performance of the SMB model" -> "Acknowledging that it might be challenging to judge the performance of the SMB model against in-situ observations"
   Changed
- 7. I71-72 Regarding "The SAM is an atmospheric phenomenon..." this seems to be a more introductory descriptive sentence and better placed a few lines earlier when the SAM is introduced

That is a good point, we suggest to move it up to L69:

".... for this reason we concentrate on its effects in this study. The SAM is an atmospheric phenomenon found across the extratropical southern

hemisphere that influences the climate over and around Antarctica (Fogt and Marshall, 2020). Marshall et al. (2017) found ......"

- 8. I88 Are these values for short-wave albedos? If so, add "shortwave" for clarity. Yes it is, from Van de Wal and Oelremans 1994 and we have added "shortwave" and the reference at line 88.
- I89-90 "Specific, for the HIRHAM5 Antarctic simulations, was that we used the Antarctic domain defined in the Coordinated Regional Climate Downscaling Experiment" -> "Specifically (or finally), for the HIRHAM5 Antarctic simulations, we used the Antarctic domain defined in the Coordinated Regional Climate Downscaling Experiment" Changed
- 10.190 12.5km resolution. This comparable size to small features often related to negative SMB: melt ponds, some glacier streams, blue ice areas. How are these handled?

See answers to comment 2

11. I100 " Despite the forcing is based on 6 hourly ..." -> " Despite the forcing being based on 6 hourly...". Also could the forcing be interpolated to 1 hr time steps, or alternatively what would the impact be of simulating the subsurface at the 6 hourly interval of the forcing?

First part: Changed,

Second part: this sentence is badly explained in the manuscript, we suggest to rephrase the sentence :

"The subsurface scheme is updated hourly by interpolating the 6 hourly forcing files to 1 hourly time steps. To ensure a smooth transition between two 6 hourly files, a linear interpolation in time between the two nearest 6 hourly files is used."

12.1101 "... model is following the ..." -> "... model follows the ..."

**Changed**

13.1122 "... which is fixed ..." -> "... which are fixed ..." OR "... the number of which are fixed ..."

Changed to: the number of which are fixed

14.1120-136 How is the bottom boundary handled? Or equivalently, how is the remainder of the mass between the lowest layer and the base of the ice sheet handled?

To clarify this, we suggest to add this, at the end of line 136: "The bottom of the lowest model layer is assumed to exchange mass and energy with an infinite layer of ice with a temperature, like in the Fixed model (Langen et al., 2015), calculated from climatological mean of the HIRHAM5 2 m temperature."

15. I157 Diagnosed snow depth? What does diagnosed refer to here?

Diagnosed snow depth refers to an estimate based on the snow concentration in each layer, calculated from the top down. It includes only snow above the first perched ice layer.

To clarify this, this could be added in line 157:

".....refreezing, diagnosed snow depth (which is an estimate based on the snow concentration in each layer), net short wave...."

16. Table 1 "Fist snow then ice" -> "First snow then ice", I think?

Yes, you are correct

17. Fig 1: The colour scale here is a little strange in the left hand panel refered to as (a) in the caption. Specifically it does not steadily get darker with a single hue, but changes part way through. This gives some bias and odd visual affects and probably makes it harder for those with colour blindness. I would suggest that the authors convert to using a standard diverging colour scheme such as Cynthia Brewer's Red-Blue scheme that can be found here: colorbrewer2.org

We have changed the color scheme accordingly (Larger figure in the paper)

18. Also please check journal guidelines for placement of caption labels (a) and (b); in their current location, they were less obvious than placed outside top-left for example.

We have checked that and make sure that the caption labels are placed consistently.

19. Fig 2: Would recommend consistent labelling of sub-figures. Here they are capitals, the last figure was lower case and the placement has changed between figures. They are refered in the text as lowercase. Also I would suggest adding a legend entry for precipitation, and chosing more distinct symbols for the TotAIS and GAIS cases.

According to the journal "Labels of panels must be included with brackets around letters being lower case (e.g. (a), (b), etc.)" so we have maked sure that the labelling are all lower case letters, consistent and referred to correctly. Legend for precipitation has been added, and the GAIS symbols have been changed too. (Larger figure in the paper)

---

## Author Response (AR2)

Hansen et al. 2021
https://tc.copernicus.org/preprints/tc-2021-69/

[Figure]

Reply to editor's comments on

**"Downscaled surface mass balance in Antarctica: impacts of subsurface processes and large-scale atmospheric circulation"**

by

Nicolaj Hansen, Peter L. Langen, Fredrik Boberg, Rene Forsberg, Sebastian B. Simonsen, Peter Thejll, Baptiste Vandecrux, and Ruth Mottram

Dear Editor Alexander Robinson,
On behalf of my co-authors and myself, I would like to thank you for your comments on our manuscript.
In the following, we provide a point-by-point answer to the issues raised by you. All issues will be followed by our suggestions for improvement to the manuscript highlighted in red.

Best regards,
Nicolaj Hansen

Hansen et al. 2021
https://tc.copernicus.org/preprints/tc-2021-69/

1. I find that the Introduction is clearly missing an explicit paragraph stating the goals of the study and what you will show. Right now, hints of this are mixed in with general introductory information, but this is confusing (see specific comments below).
*We have written a final paragraph dedicated to this, at the end of the introduction:*
*"The aims of this study are thus; to estimate present day Antarctic SMB using our subsurface model forced with the RCM HIRHAM5, compare and evaluate the two subsurface model versions against each other and in-situ data. Furthermore, we estimate the MB, using our modelled SMB results combined with discharge values, and compare it with GRACE, and finally we investigate the relationship between the SAM and the SMB. This is done in the following structure; first a method section, where the RCM HIRHAM5, the two subsurface models and their set-up are described. Followed by a result section, where the modelled SMB results are shown, including evaluation against in-situ measurements of SMB, firn temperature and density. Then the discussion, where, besides the results, the MB is estimated and evaluated against GRACE data is discussed and also the influence of SAM on SMB. Finally, a conclusion in the end."*

Specific comments:

2. L23: contributes to sea level rise by 0.3±0.16 mm yr-1 => contributes 0.3±0.16 mm yr-1 to sea-level rise
*Changed*

3. L26: in induce ice sheet dynamical instability => in inducing ice-sheet dynamic instability
*Changed*

4. L31: Rephrase, as you have not yet introduced any particular modeling approach. Perhaps simply delete and mention it elsewhere.
*As the comment on blowing snow came from one of the reviewers we believe it should stay. However, your point about modeling is very true. So the sentence has been rephrased to:*
*"...However, blowing snow is not taken into consideration in this study, so the SMB is defined here as:..."*

5. L33: higher altitudes => higher altitudes,
*Changed*

6. L37: dynamical => dynamic [and elsewhere]
*Changed*

[Figure]

7. L51: Again, here it is strange to mention what you do with HIRHAM, as you have not yet introduced that you have a model, or what you plan to do with it. I would remove this sentence.
We have added a sentence in L37, see below
*"…Here we focus on the SMB component of the mass balance, to pin-point the immediate forcing to ice sheet dynamic instability. To estimate the SMB we use an atmospheric Regional Climate Model (RCM) to force a subsurface model, which outputs the SMB.*
*Regional Climate Models are most often used…."*
By adding the new sentence from, we believe that this sentence could still stand, by adding "*RCM*" before HIRHAM5

8. L56: "we also compare our model results" <= Again, what model results?
We have added SMB in that sentence:
*"...we also compare our modelled SMB results…"*

9. L73: sea ice extent => sea-ice extent
Changed

10. L84: global circulation models, for full => global circulation models. For a full
Changed

11. L103: physically state => physical state
Changed

12. L110-113: Run-on sentence. Please separate.
Done

13. L114: on the layer's => on a layer's
Changed

14. L117: Delete "However"
Done

15. L120: This phrase doesn't seem to make sense: "that mass is advected through layers of fixed mass". How can mass be advected through layers of fixed mass? Please revise.
As mass is added on top of the subsurface model, the scheme advects mass downward to ensure the constant w.eq. layer thicknesses. To make it more clear we have removed the sentence "that mass is advected through layers of fixed mass". It is rewritten to this:
*"This fixed model implies an Eulerian framework, meaning that when snowfall occurs at the surface, it is added to the first layer and an equal mass from that layer is shifted to the underlying layer. The same goes for each layer in the model column."*

16. L127: dynamical => dynamic
Changed

17. L166-167: 3d => 3D; 2d => 2D [and elsewhere]
    Changed

18. L231,L233: data set => dataset [and elsewhere]
    Changed

19. Fig. 3, caption: the charge => the change [?]
    Corrected

20. L257: overestimates => overestimate
    Corrected

21. L258: underestimates => underestimate
    Corrected

22. L261: are consistent => is consistent
    Corrected

---

## Author Response (AR3)

Hansen et al. 2021
https://tc.copernicus.org/preprints/tc-2021-69/

[Figure]

Reply to editor's comments on

**"Downscaled surface mass balance in Antarctica: impacts of subsurface processes and large-scale atmospheric circulation"**

by

Nicolaj Hansen, Peter L. Langen, Fredrik Boberg, Rene Forsberg, Sebastian B. Simonsen, Peter Thejll, Baptiste Vandecrux, and Ruth Mottram

Dear Editor Alexander Robinson,
On behalf of my co-authors and myself, I would like to thank you for your comments on our manuscript.
We have corrected all six comments.

Best regards,
Nicolaj Hansen

Hansen et al. 2021
https://tc.copernicus.org/preprints/tc-2021-69/

1. L83: present day Antarctic SMB => present-day Antarctic SMB
   Corrected

2. L84: the two subsurface model versions => two subsurface model versions
   Corrected

3. L85: GRACE, and finally => GRACE. Finally,
   Corrected

4. L86: first a method section => first, the methods are presented,
   Corrected

5. L87: Followed by a result section => This is followed by the results
   Corrected

6. L89:Then the discussion, where, besides the results, the MB is estimated and
   evaluated against GRACE data is discussed and also the influence of SAM on SMB.
   Finally, a conclusion in the end.
   => Finally, the MB is estimated and evaluated against GRACE data, and we discuss
   the influence of SAM on SMB, followed by the conclusions.
   Corrected